# LEARNING IN TRANSFORMERS UNDER SPECTRAL CONSTRAINTS

**Md Rifat Arefin[1, 2, 3]**,* **Ravid Shwartz-Ziv[1], Ernie Chang[1], Chinnadhurai Sankar[1], Rylan Conway[1],**
**Aristide Baratin[2, 3], Adithya Sagar[1], Patrick Huber[1]**
[1]Meta        [2]Mila-Quebec AI Institute        [3]University of Montreal        [4]Samsung
{rifat.arefin}@mila.quebec

## ABSTRACT

Training large Transformers is often bottlenecked by early optimization dynamics rather than model capacity. In this work, we identify a concrete spectral pathology that emerges early in training and show that it can be mitigated with minimal, targeted intervention. Specifically, we show that a small number of weight matrices—namely the attention output projection and the MLP down-projection—develop severe spectral spikiness characterized by rapid growth of the top singular value relative to the bulk spectrum. This induces ill-conditioning, distorts gradient flow, and slows convergence in language model pretraining. We demonstrate that geometry-aware optimization (Muon) suppresses this pathology by implicitly controlling matrix spectra, while standard AdamW lacks such regulation. Crucially, we show that targeted, early-phase spectral stabilization applied only to these matrices further improves Muon's performance as well as benefit AdamW. Our results identify spectral conditioning on certain layers as a central optimization bottleneck in Transformers and show that minimal, localized geometric control is sufficient to substantially accelerate learning.

## 1 INTRODUCTION

Transformer pretraining is unusually sensitive to optimization choices, and small differences in *early* dynamics can yield large gaps in speed and final loss (Pan & Li, 2023; Semenov et al., 2025). A central geometric driver is *early spectral anisotropy*: intermediate token representations—and the weight matrices they feed—can become low stable/effective rank, causing gradients and optimizer updates to concentrate into a low-dimensional subspace. When these *effective updates* remain temporally aligned, successive steps repeatedly reinforce the same correlation patterns in $W^\top W$, slowing optimization.

This phenomenon is not purely a training artifact. Attention mixing induces spectral bias already at initialization, with depth amplifying contraction toward a few dominant modes (Dong et al., 2021; Saada et al., 2025), and rectifying/gated MLP parameterizations can create additional low-rank interfaces inside each block (Davis & Drusvyatskiy, 2025). However, existing analyses are often layer-agnostic, leaving unclear where these pathologies localize in modern blocks and how they interact with practical optimizers.

We address this gap through a localized early-phase lens: *runaway correlated updates*. Empirically, we show that the strongest early collapse concentrates at the attention value/output interface and the MLP contraction—specifically $W_V$, $W_O$, and $W_{\text{down}}$—while most other projections remain comparatively well-conditioned. We use optimizer geometry as a probe: Muon's geometry-normalized updates mitigate the same localized failure mode and accelerate training under identical architectures (Newhouse et al., 2025), suggesting that early optimization is bottlenecked by localized spectral geometry rather than capacity or data difficulty. This motivates a minimal intervention: can we directly suppress early correlation accumulation in $\{W_V, W_O, W_{\text{down}}\}$ and recover gains without changing the optimizer? Concretely, we ask *where and when* spectral pathologies arise inside modern Transformers, and *how minimally* they can be disrupted to accelerate pretraining.

---

*Work done during an internship at Meta.

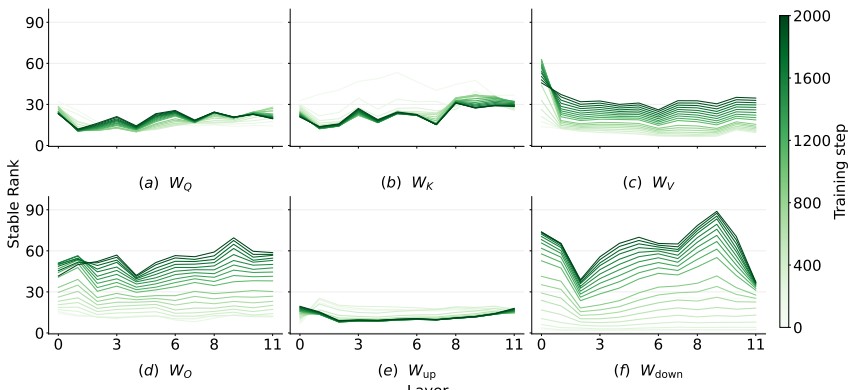

Figure 1: **Stable-rank localization** under **AdamW + ReLU$^2$** (no warmup): early collapse concentrates at $W_V$, $W_O$, and $W_{\text{down}}$, while other projections remain comparatively well-conditioned.

Our contributions are as follows:

- **Localization.** We observe that early effective-/stable-rank collapse is sharply localized to $W_V$, $W_O$, and $W_{\text{down}}$.

- **Mechanism.** We propose a runaway correlated-update account: low-dimensional, temporally persistent *effective updates* amplify off-diagonal Gram correlations in these matrices.

- **Intervention.** We show that an *early-only* off-diagonal Gram penalty applied solely to $\{W_V, W_O, W_{\text{down}}\}$ improves optimization and validation loss.

## 2 RELATED WORK

Transformers exhibit strong, largely *architectural* spectral anisotropy: attention-only stacks can rapidly lose rank with depth even near initialization (Dong et al., 2021; Barbero et al., 2024; Wu et al., 2024), and Markov-style analyses predict a spectral gap that concentrates mass into a dominant Perron–Frobenius–like mode as width/sequence length grow (Saada et al., 2025). MLP nonlinearities can further sharpen this bias, as rectifying or non-centered activations (e.g., ReLU$^2$ and some gated MLPs) concentrate energy into few directions and reduce effective rank (Davis & Drusvyatskiy, 2025; Bernstein & Newhouse, 2024).

On the optimization side, geometry-aware updates can be critical in such regimes: spectral gradient descent helps under low-stable-rank / high nuclear-to-Frobenius conditions (Davis & Drusvyatskiy, 2025), and Muon improves stability by reducing sensitivity to gradient scale (Jordan et al., 2024; Newhouse et al., 2025), with related scale-invariant/manifold-constrained updates in representation learning (Yang & Hu, 2021). These gains appear localized—Muon matters most on value/output and MLP parameters (Wang et al., 2025)—matching our finding that $W_O$ and $W_{\text{down}}$ are early accumulation points for rank collapse and correlation growth. Finally, whereas early stabilization often uses warmup or global norm/Lipschitz controls (Goyal et al., 2017; Kalra & Barkeshli, 2024; Sedghi et al., 2019; Newhouse et al., 2025), we study a minimal transient alternative: an early-only, localized off-diagonal Gram penalty that interrupts correlation accumulation without enforcing global isometry.

## 3 PRELIMINARIES AND DIAGNOSTICS

We study a decoder-only Transformer (details in A.1) (length $T$, width $d$) and focus on the attention value/output and MLP down-projection matrices $W_V, W_O \in \mathbb{R}^{d \times d}$ and $W_{\text{down}} \in \mathbb{R}^{d \times d_{\text{ff}}}$. As a conditioning proxy we use stable rank $\text{sr}(M) := \|M\|_F^2 / \|M\|_2^2$ (small values indicate domination by a few singular directions). For any linear projection $Y = XW^\top$, gradients factorize as $\nabla_W \ell = (\partial \ell / \partial Y)^\top X$, so low-rank/an-isotropic token matrices bottleneck gradient (and hence update) rank. Full diagnostic definitions (SNR/TopShare/temporal alignment) are in Appendix B.

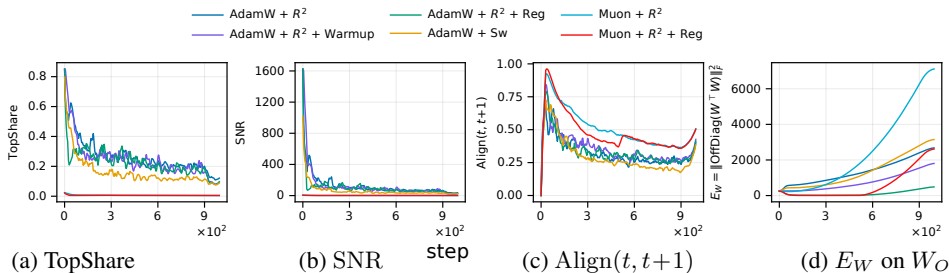

(a) TopShare  (b) SNR  (c) $\mathrm{Align}(t, t+1)$  (d) $E_W$ on $W_O$

Figure 2: **Update-geometry and Gram-correlation at $W_O$ (layer-averaged).** Other matrices are deferred to Appendix F.1.

## 4 MECHANISM: RUNAWAY CORRELATED UPDATES AND EARLY SPECTRAL COLLAPSE

We give a minimal account of why early rank collapse localizes to $\{W_V, W_O, W_{\mathrm{down}}\}$ and why a small off-diagonal Gram penalty helps. Our key abstraction is the *effective update* $\Delta W_{\ell,t}$ produced by the optimizer (AdamW-preconditioned or Muon-orthogonalized), rather than the raw gradient. Formal definitions and proofs are organized in Appendix D (see the *Roadmap* in App. D).

**Low-rank inputs $\Rightarrow$ low-rank effective updates (setup).** In an early window, token matrices feeding the attention value/output interface and the MLP contraction are low stable/effective rank (Assumption in App. C.1). Since linear projections satisfy $\nabla_W \ell = \Delta_Y^\top X$ (Lemma D.1) and singular values of products are bottlenecked by either factor (Lemma D.2), gradients—and hence effective updates $\Delta W_{\ell,t}$—are typically low-dimensional for $W_V, W_O, W_{\mathrm{down}}$.

**Runaway regime (informal).** We say layer $\ell$ is in a *runaway correlated-update* regime over an early window if (i) its incoming token matrix is low-rank, (ii) $\Delta W_{\ell,t}$ is low-rank, and (iii) updates have nontrivial coherence and temporal alignment; see Appendix C for formal statements and diagnostics.

**Gram identity and two accumulation channels.** Let $G_t := W_t^\top W_t$ and track correlation via $E_t := \|\mathrm{OffDiag}(G_t)\|_F^2$. For any optimizer-induced update $W_{t+1} = W_t + \Delta W_t$,

$$G_{t+1} - G_t = \underbrace{W_t^\top \Delta W_t + \Delta W_t^\top W_t}_{\Delta G_t^{(1)}} + \underbrace{\Delta W_t^\top \Delta W_t}_{\Delta G_t^{(2)} \succeq 0}. \tag{1}$$

This decomposition is exact (Lemma D.3). (**AdamW-like**) If $\Delta W_t$ is anisotropic, the PSD term $\Delta G_t^{(2)}$ injects mass into a stable direction, increasing off-diagonal energy (formal rank-one evolution in Theorem D.4). (**Muon-like**) Muon approximately orthogonalizes matrix updates, making $\Delta W_t^\top \Delta W_t \approx c_t I$ so $\mathrm{OffDiag}(\Delta G_t^{(2)}) \approx 0$ (Theorem D.8); then off-diagonal growth is dominated by cross-term drift $\Delta G_t^{(1)}$, amplified by temporal persistence.

**Targeted Gram-offdiag regularization.** We use the localized penalty

$$R_0(W) := \|\mathrm{OffDiag}(W_O^\top W_O)\|_F^2 \tag{2}$$

applied only to $\{W_V, W_O, W_{\mathrm{down}}\}$ for early steps. Its gradient (Lemmas D.5–D.6) induces a restoring drift that decreases off-diagonal energy (Theorem D.7), disrupting both accumulation channels.

**Takeaway.** Early operator-induced low-rankness creates low-dimensional, temporally persistent *effective updates* in a small set of matrices; the Gram identity links these updates to off-diagonal correlation growth under AdamW (quadratic injection) and Muon (cross-term drift), and explains why a localized off-diagonal Gram penalty breaks the feedback loop.

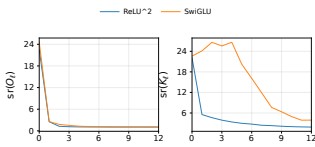
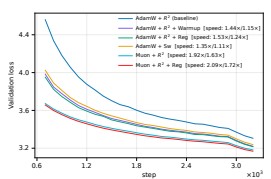
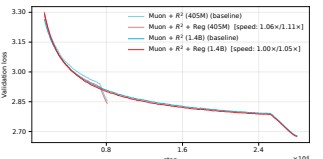

(a) **Init bottlenecks.** Stable/effective rank vs. depth for post-attn $O_\ell$ and expanded MLP activations $K_\ell$ at initialization.

(b) **162M ($\text{ReLU}^2(R^2)$).** Val. loss vs. steps; speedup at Losses $L^\star = \{3.50, 3.35\}$. Muon+$R_0$ (Reg) best ($\approx 2.09 \times / 1.72\times$).

(c) **Scale ($\text{ReLU}^2$ ($R^2$).** Muon vs. Muon+$R_0$ (Reg) at 405M/1.4B ($R_0$ for first 10%). Speedup at Losses $L^\star_{405} = \{3.25, 3.0\}$, $L^\star_{1.4B} = \{3.10, 2.85\}$.

Figure 3: Left: low-rank bottlenecks are present at initialization. Middle/right: Muon accelerates optimization and the targeted Gram-offdiag regularizer $R_0$ provides gains at 162M and at scale.

## 5   EMPIRICAL LOCALIZATION OF EARLY SPECTRAL PATHOLOGY

We localize when and where spectral degeneracy arises during pretraining and relate it to structured update dynamics. Across settings, the only projections that exhibit a pronounced *early* collapse in effective dimensionality are $W_V$, $W_O$, and $W_{\text{down}}$ (§3). **Low-rank geometry appears before learning:** at initialization, representations entering the attention output and MLP down projections are already anisotropic (Fig. 3a), with sharp rank drops (i) immediately after attention mixing (post-attention $O$) and (ii) at the MLP expansion–contraction interface (expanded activations $K$ feeding $W_{\text{down}}$); the effect is strongest for $\text{ReLU}^2$, while SwiGLU preserves higher rank in the expanded space but still contracts to a lower-dimensional subspace. **Early collapse is localized and two-phase:** tracking stable rank under AdamW+$\text{ReLU}^2$ (no warmup) shows a sharp early collapse concentrated in $\{W_V, W_O, W_{\text{down}}\}$ followed by partial recovery, whereas most other projections remain well-conditioned early and become more anisotropic later (Fig. 1). **Update-geometry signatures precede Gram correlation growth:** for $W_O$, early AdamW+$\text{ReLU}^2$ exhibits elevated $\text{TopShare}_O$ and $\text{SNR}_O$ alongside rapid growth of $\|\text{OffDiag}(W_O^\top W_O)\|_F^2$ (Fig. 2), consistent with low-dimensional coherent updates; Muon shows higher temporal alignment but substantially lower $\text{TopShare}_O/\text{SNR}_O$, and adding the targeted Gram off-diagonal regularizer suppresses early $\text{TopShare}_O/\text{SNR}_O$ spikes and reduces off-diagonal growth (Appendix F.1). **Takeaway:** early spectral pathology is sharply localized to $\{W_V, W_O, W_{\text{down}}\}$, co-occurs with low-rank token representations, and is accompanied by temporally persistent, low-dimensional updates—motivating the mechanism in §4.

## 6   EXPERIMENTAL SETUP

We study decoder-only LM pretraining to test whether early, *localized* spectral spikiness slows optimization and whether it is mitigated by geometry-aware updates (Muon) or a minimal early-only regularizer. We train standard pre-norm Transformers (RoPE; causal attention+MLP) from scratch on FineWeb (Penedo et al., 2024) using the pipeline of Ahn et al. (2025) with the GPT-2 tokenizer; default $B = T = 1024$ (so $BT$ tokens/step). We compare $\text{ReLU}^2$ (So et al., 2021) vs. SwiGLU (Shazeer, 2020; Touvron et al., 2023) across 160M/405M/1.4B models, and AdamW (Loshchilov & Hutter, 2017) vs. Muon (Jordan et al., 2024). All interventions target only $W_V$, $W_O$, and $W_{\text{down}}$ (§3). To suppress early correlation accumulation without changing the optimizer, we add for $t \leq t_{\text{reg}}$ an off-diagonal Gram penalty $R_0(W)$ layerwise to these matrices, $\mathcal{L} = \mathcal{L}_{\text{NTP}} + \lambda \sum_{\ell=1}^{L} \big( R_0(W_{V,\ell}) + R_0(W_{O,\ell}) + R_0(W_{\text{down},\ell}) \big)$, and set $\lambda = 0$ afterward. Full hyperparameters are in Appendix E.2–E.4.

**Optimization speed and scaling results.**   Under $\text{ReLU}^2$, Muon substantially accelerates optimization relative to AdamW at 162M, and the targeted Gram-offdiag regularizer ($R_0$ on $\{W_V, W_O, W_{\text{down}}\}$ for the first 10% of steps) provides an additional improvement, with Muon+$R_0$ achieving the fastest trajectories (Fig. 3b). At larger scales (405M, 1.4B), adding $R_0$ to Muon remains beneficial but the marginal gain shrinks (Fig. 3c), consistent with Muon already mitigating much of the early conditioning failure while $R_0$ supplies a small residual correction. Additional SwiGLU results are deferred to Appendix F.2.

## 7 Conclusion and Limitations

We study the early phase of Transformer pretraining through a spectral-geometry lens and find that effective-rank collapse is sharply *localized*, arising first and most strongly at the attention value/output interface and the MLP contraction (concentrating in $W_V$, $W_O$, and $W_{\text{down}}$). These matrices exhibit a runaway regime in which operator-induced anisotropy and temporally persistent, low-dimensional *effective updates* reinforce correlation structure across steps. Motivated by this mechanism, we introduce a minimal intervention—an *early-only* off-diagonal Gram penalty applied exclusively to $\{W_V, W_O, W_{\text{down}}\}$ and disabled after the first 10% of training—that improves validation loss under both AdamW and Muon, indicating benefits complementary to optimizer-level geometry normalization. A key limitation is that our intervention is intentionally *transient* (fixed 10% on/off): an adaptive schedule based on online diagnostics (e.g., TopShare/SNR or Gram growth), understanding whether/when to re-activate it in later phases (especially under LR cooldown), and extending the approach to late-phase correlation growth in other projections without harming generalization remain open; we also leave a fuller accounting of compute overhead vs. speedup and transferability across tokenizers, domains, and longer contexts to future work.

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

# 8 APPENDIX

# A MODEL, NOTATION, AND BLOCK DEFINITIONS

## A.1 DECODER-ONLY TRANSFORMERS

We consider a decoder-only Transformer with $L$ layers, each composed of a self-attention block followed by an MLP block. The input is a sequence of $T$ tokens with embedding dimension $d$, collected into a matrix $X_0 \in \mathbb{R}^{T \times d}$.

For each layer $\ell = 1, \ldots, L$, we write $X_\ell \in \mathbb{R}^{T \times d}$ for the layer output (and input to the next layer), with rows $x_{\ell,t} \in \mathbb{R}^d$ denoting the representation of token $t$. Within layer $\ell$, we denote the outputs of the self-attention and MLP sublayers by $X_\ell^{\text{Attn}} \in \mathbb{R}^{T \times d}$ and $X_\ell^{\text{MLP}} \in \mathbb{R}^{T \times d}$, respectively; in particular, $X_\ell^{\text{Attn}}$ is the attention output matrix $O_\ell$ used elsewhere in the paper.

**Self-Attention Block.** In layer $\ell$, the self-attention[1] block takes $X_{\ell-1} \in \mathbb{R}^{T \times d}$ as input and uses projection matrices $W_Q^{(\ell)}$, $W_K^{(\ell)}$, $W_V^{(\ell)}$, $W_O^{(\ell)} \in \mathbb{R}^{d \times d}$.

We form queries, keys, and values by

$$Q_\ell = X_{\ell-1} W_Q^{(\ell)}, \quad K_\ell = X_{\ell-1} W_K^{(\ell)}, \quad V_\ell = X_{\ell-1} W_V^{(\ell)} \in \mathbb{R}^{T \times d}. \tag{3}$$

The (causal) attention logits and weights are

$$S_\ell = \frac{1}{\sqrt{d}} Q_\ell K_\ell^\top \in \mathbb{R}^{T \times T}, \qquad A_\ell = \mathrm{softmax}(S_\ell) \in \mathbb{R}^{T \times T}, \tag{4}$$

where the softmax is applied row-wise and $S_{\ell,t,s}$ is set to $-\infty$ for $s > t$ to enforce the causal mask.

The attention output and its output projection are

$$U_\ell = A_\ell V_\ell \in \mathbb{R}^{T \times d}, \qquad X_\ell^{\mathrm{Attn}} = U_\ell W_O^{(\ell)} \in \mathbb{R}^{T \times d}. \tag{5}$$

We write $x_{\ell,t}^{\mathrm{Attn}}$ for the $t$-th row of $X_\ell^{\mathrm{Attn}}$.

**MLP block.** In layer $\ell$, let $X_\ell^{\mathrm{Attn}} \in \mathbb{R}^{T \times d}$ denote the $T$ token representations (width $d$) entering the feedforward (MLP) block. With hidden width $m$, the MLP forms post-activation hidden states $K_\ell \in \mathbb{R}^{T \times m}$ and returns to the model dimension via the down-projection $W_{\mathrm{down}}^{(\ell)} \in \mathbb{R}^{m \times d}$:

$$X_\ell^{\mathrm{MLP}} = K_\ell W_{\mathrm{down}}^{(\ell)} \in \mathbb{R}^{T \times d}, \tag{6}$$

producing one output vector in $\mathbb{R}^d$ per token.

We consider two common parameterizations for $K_\ell$.

*Non-gated (additive) MLP.* With an up-projection $W_{\mathrm{up}}^{(\ell)} \in \mathbb{R}^{d \times m}$ and elementwise nonlinearity $\phi$ (e.g., ReLU, GELU, ReLU$^2$),

$$K_\ell^{\mathrm{non\text{-}gate}} = \phi\big(X_\ell^{\mathrm{Attn}} W_{\mathrm{up}}^{(\ell)}\big) \in \mathbb{R}^{T \times m}. \tag{7}$$

*Gated (SwiGLU-style) MLP.* With two up-projections $W_u^{(\ell)}, W_g^{(\ell)} \in \mathbb{R}^{d \times m}$,

$$K_\ell^{\mathrm{gate}} = \phi\big(X_\ell^{\mathrm{Attn}} W_u^{(\ell)}\big) \odot \big(X_\ell^{\mathrm{Attn}} W_g^{(\ell)}\big) \in \mathbb{R}^{T \times m}, \tag{8}$$

where $\odot$ denotes elementwise (Hadamard) product. In practice, $\phi$ is variant-dependent (commonly SiLU for SwiGLU, and ReLU/GELU/ReLU$^2$ for non-gated MLPs).

**Combined Block (pre-norm).** Putting the attention and MLP sublayers together, a pre-norm decoder-only block in layer $\ell$ maps $X_{\ell-1} \in \mathbb{R}^{T \times d}$ to $X_\ell \in \mathbb{R}^{T \times d}$ via

$$\tilde{X}_\ell^{\mathrm{Attn}} = \mathrm{LN}\big(X_{\ell-1}\big), \tag{9}$$

$$X_\ell^{\mathrm{Attn}} = X_{\ell-1} + \mathrm{Attn}_\ell\big(\tilde{X}_\ell^{\mathrm{Attn}}\big), \tag{10}$$

$$\tilde{X}_\ell^{\mathrm{MLP}} = \mathrm{LN}\big(X_\ell^{\mathrm{Attn}}\big), \tag{11}$$

$$X_\ell = X_\ell^{\mathrm{Attn}} + \mathrm{MLP}^{(\ell)}\big(\tilde{X}_\ell^{\mathrm{MLP}}\big), \tag{12}$$

where $\mathrm{Attn}_\ell(\cdot)$ is the single-head causal self-attention map defined above and $\mathrm{MLP}^{(\ell)}(\cdot)$ is the MLP block from the previous subsection.

## A.2 ADAM VS. MUON

**Adam** Let $\theta$ denote parameters, $g_t = \nabla_\theta \mathcal{L}_t$. AdamW maintains first/second moments

$$m_t = \beta_1 m_{t-1} + (1 - \beta_1) g_t, \quad v_t = \beta_2 v_{t-1} + (1 - \beta_2) g_t \odot g_t,$$

with bias corrections $\hat{m}_t = m_t / (1 - \beta_1^t)$ and $\hat{v}_t = v_t / (1 - \beta_2^t)$, and applies a *decoupled* weight decay $\lambda$:

$$\theta_{t+1} \leftarrow (1 - \eta\lambda)\,\theta_t - \eta\,\frac{\hat{m}_t}{\sqrt{\hat{v}_t} + \varepsilon}.$$

AdamW is elementwise adaptive.

---

[1]For simplicity we use single head here

**Muon** For each *matrix* parameter $W \in \mathbb{R}^{d_\text{out} \times d_\text{in}}$ (e.g., linear, conv kernels as flattened matrices, attention $Q, K, V, O$), Muon first forms a momentum update $M_t = \mu M_{t-1} + (1 - \mu) G_t$ with $G_t = \partial \mathcal{L} / \partial W$, then *orthogonalizes* that update by approximately projecting $M_t$ to the nearest semi-orthogonal matrix:

$$\text{Ortho}(M) \approx \underset{O^\top O = I \text{ or } OO^\top = I}{\arg\min} \|O - M\|_F = UV^\top \text{ if } M = U\Sigma V^\top.$$

Instead of computing an SVD, Muon applies $K$ steps of a Newton–Schulz–style polynomial to $X_0 = M_t / \|M_t\|_F$:

$$X_{k+1} = a X_k + b(X_k X_k^\top) X_k + c(X_k X_k^\top)^2 X_k,$$

yielding $X_K \approx \text{Ortho}(M_t)$. The final update (with RMS–to–spectral normalization) is

$$W \leftarrow W - \eta \sqrt{\tfrac{d_\text{out}}{d_\text{in}}} X_K.$$

Non-matrix parameters (biases, LayerNorm scales) and typically embeddings/heads are optimized with AdamW.

## B DIAGNOSTIC QUANTITIES AND ESTIMATION DETAILS

**Stable rank.** For a matrix $M$, $\text{sr}(M) := \|M\|_F^2 / \|M\|_2^2$.

**Outer-product gradients (linear projections).**

**Lemma B.1** (Gradient factorization). *Let $Y = XW^\top$ with $X \in \mathbb{R}^{T \times d_\text{in}}$ and $W \in \mathbb{R}^{d_\text{out} \times d_\text{in}}$. If $\Delta_Y = \partial \ell / \partial Y$, then $\nabla_W \ell = \Delta_Y^\top X$.*

**Gradient coherence and concentration.** Let $g_\ell(x) = \text{vec}(\nabla_{W_\ell} \ell(x))$, $\mu_\ell = \mathbb{E}[g_\ell(x)]$, and $\Sigma_\ell = \text{Cov}(g_\ell(x))$. Define

$$\text{SNR}_\ell = \frac{\|\mu_\ell\|^2}{\text{tr}(\Sigma_\ell)}, \qquad \mathcal{A}_\ell = \frac{\|\mu_\ell\|^2}{\mathbb{E}\|g_\ell(x)\|^2}, \qquad \text{TopShare}_\ell = \frac{\lambda_1(\Sigma_\ell)}{\text{tr}(\Sigma_\ell)}. \tag{13}$$

For temporal persistence, let $\bar{g}_{\ell,t} = \mathbb{E}[g_{\ell,t}(x)]$ and define

$$\text{Align}_\ell(t, t+1) = \frac{\langle \bar{g}_{\ell,t}, \bar{g}_{\ell,t+1} \rangle}{\|\bar{g}_{\ell,t}\| \|\bar{g}_{\ell,t+1}\|}. \tag{14}$$

## C RUNAWAY REGIMES AND LOCALIZATION ASSUMPTIONS

**Assumption C.1** (Early operator-induced anisotropy). In an early training window, token matrices entering certain projections (e.g. post-attention values and post-MLP activations) have low stable/effective rank, as predicted for attention and rectifying/gated operators and observed empirically (Fig. 1).

**Lemma C.2** (Gradient spectrum bottleneck). *Let $g = \Delta^\top U$ with conformable $\Delta, U$. For all $k$,*

$$\sigma_k(g) \le \sigma_1(\Delta) \sigma_k(U), \qquad \sigma_k(g) \le \sigma_k(\Delta) \sigma_1(U).$$

**Definition C.3** (Runaway correlated *update* regime). A layer $\ell$ is in a runaway correlated update regime on $\mathcal{T}$ if: (i) incoming token matrices are low stable/effective rank for most $t \in \mathcal{T}$; (ii) $\Delta W_{\ell,t}$ has low stable/effective rank for most $t \in \mathcal{T}$; (iii) update SNR is non-negligible (Appendix B); and (iv) successive updates are temporally aligned, e.g. $\langle \Delta W_{\ell,t}, \Delta W_{\ell,t+1} \rangle_F > 0$ for most $t \in \mathcal{T}$.

## D PROOFS FOR SECTION 4

### D.1 ROADMAP

Section 4 uses: (i) outer-product gradients (Lemma D.1); (ii) a product singular-value bound (Lemma D.2); (iii) the Gram identity and its rank-one specialization (Lemma D.3, Theorem D.4); (iv) gradients and descent drift for off-diagonal Gram penalties (Lemmas D.5–D.6, Theorem D.7); (v) the AdamW/Muon channel split (Theorem D.8).

## D.2   NOTATION AND BASIC OPERATORS

Let $W \in \mathbb{R}^{d_{\text{out}} \times d_{\text{in}}}$ and define the (input-side) Gram matrix

$$G(W) := W^\top W \in \mathbb{R}^{d_{\text{in}} \times d_{\text{in}}}.$$

For any $M \in \mathbb{R}^{d \times d}$ define the diagonal and off-diagonal projections entrywise:

$$\text{Diag}(M)_{ij} := \begin{cases} M_{ii}, & i = j, \\ 0, & i \neq j, \end{cases} \qquad \text{OffDiag}(M)_{ij} := \begin{cases} 0, & i = j, \\ M_{ij}, & i \neq j. \end{cases}$$

Equivalently, $\text{OffDiag}(M) = M - \text{Diag}(M)$. We use the Frobenius inner product $\langle A, B \rangle := \text{tr}(A^\top B)$ and $\|A\|_F^2 = \langle A, A \rangle$.

Throughout, $g$ denotes a raw gradient w.r.t. $W$ (per-sample or mean), and $\Delta W$ denotes the *effective update* applied by the optimizer at that step.

## D.3   TOKEN-MATRIX GRADIENT FACTORIZATION

**Lemma D.1** (Token-matrix gradient factorization). *Let $Y = XW^\top$, where $X \in \mathbb{R}^{T \times d_{\text{in}}}$ and $W \in \mathbb{R}^{d_{\text{out}} \times d_{\text{in}}}$. Let $\Delta_Y := \partial \ell / \partial Y \in \mathbb{R}^{T \times d_{\text{out}}}$. Then*

$$\nabla_W \ell = \Delta_Y^\top X.$$

*Proof.* Write $Y_{t,o} = \sum_{i=1}^{d_{\text{in}}} X_{t,i} W_{o,i}$. Then

$$\frac{\partial \ell}{\partial W_{o,i}} = \sum_{t=1}^T \frac{\partial \ell}{\partial Y_{t,o}} \frac{\partial Y_{t,o}}{\partial W_{o,i}} = \sum_{t=1}^T (\Delta_Y)_{t,o} X_{t,i},$$

which is exactly the $(o, i)$ entry of $\Delta_Y^\top X$.    □

## D.4   SINGULAR-VALUE BOTTLENECK

**Lemma D.2** (Product singular-value bound). *For conformable matrices $A, B$ and any $k \geq 1$,*

$$\sigma_k(AB) \leq \sigma_1(A)\, \sigma_k(B), \qquad \sigma_k(AB) \leq \sigma_k(A)\, \sigma_1(B).$$

*Proof.* We prove $\sigma_k(AB) \leq \sigma_1(A)\sigma_k(B)$; the other inequality follows by applying the same argument to $(AB)^\top = B^\top A^\top$. Using the variational characterization,

$$\sigma_k(AB) = \min_{\dim(\mathcal{S})=k-1} \max_{\substack{x \perp \mathcal{S} \\ \|x\|=1}} \|ABx\|.$$

Fix any $(k-1)$-dimensional subspace $\mathcal{S}$. For $x \perp \mathcal{S}$ with $\|x\| = 1$,

$$\|ABx\| \leq \|A\|_2 \|Bx\| = \sigma_1(A) \|Bx\|.$$

Taking $\max_{x \perp \mathcal{S}, \|x\|=1}$ and then minimizing over $\mathcal{S}$ yields

$$\sigma_k(AB) \leq \sigma_1(A)\, \sigma_k(B).$$

   □

**Instantiation.**    With $g = \Delta^\top U$, apply Lemma D.2 to $(A, B) = (\Delta^\top, U)$ and to $(A, B) = (U, \Delta^\top)$ to obtain Lemma C.2 in the main text.

## D.5   EXACT GRAM UPDATE DECOMPOSITION

**Lemma D.3** (Exact Gram update). *Let $G_t := W_t^\top W_t$ and $W_{t+1} = W_t + \Delta W_t$. Then*

$$G_{t+1} - G_t = W_t^\top \Delta W_t + \Delta W_t^\top W_t + \Delta W_t^\top \Delta W_t.$$

*Proof.* Expand

$$G_{t+1} = (W_t + \Delta W_t)^\top (W_t + \Delta W_t) = W_t^\top W_t + W_t^\top \Delta W_t + \Delta W_t^\top W_t + \Delta W_t^\top \Delta W_t.$$

   □

## D.6   RANK-ONE UPDATE GRAM EVOLUTION

**Theorem D.4** (Rank-one update Gram evolution). *Let $W_{t+1} = W_t + \Delta W_t$ and $G_t = W_t^\top W_t$. If $\Delta W_t = -\eta_t a_t b^\top$ with $\|b\| = 1$, then*

$$G_{t+1} = G_t - \eta_t\big(W_t^\top a_t b^\top + b a_t^\top W_t\big) + \eta_t^2 \|a_t\|^2\, bb^\top.$$

*In particular, the $\eta_t^2$ term is PSD and rank-one in direction $b$.*

*Proof.* Apply Lemma D.3 with $\Delta W_t = -\eta_t a_t b^\top$:

$$G_{t+1} - G_t = -\eta_t(W_t^\top a_t b^\top + b a_t^\top W_t) + \eta_t^2 (a_t b^\top)^\top (a_t b^\top).$$

Since $(a_t b^\top)^\top (a_t b^\top) = b(a_t^\top a_t)b^\top = \|a_t\|^2\, bb^\top$, the claim follows.    $\square$

**Interpretation**    The $\eta_t^2$ term is PSD and concentrates mass in input-side direction $b$, capturing the "quadratic injection" channel.

## D.7   GRADIENTS OF OFF-DIAGONAL GRAM PENALTIES

We provide both the squared energy gradient (often convenient in derivations) and the unsquared norm used in the main text.

**Lemma D.5** (Gradient of squared off-diagonal Gram energy). *Let $E(W) := \|\mathrm{OffDiag}(W^\top W)\|_F^2$ and $C := \mathrm{OffDiag}(W^\top W)$. Then*

$$\nabla_W E(W) = 4\, W\, C.$$

*Proof.* Let $G = W^\top W$ so $dG = W^\top dW + dW^\top W$. Since OffDiag is linear,

$$dE = 2\langle C,\, \mathrm{OffDiag}(dG)\rangle.$$

Because $C$ has zero diagonal, $\langle C, \mathrm{Diag}(dG)\rangle = 0$, hence $\langle C, \mathrm{OffDiag}(dG)\rangle = \langle C, dG\rangle$. Thus

$$dE = 2\langle C, W^\top dW + dW^\top W\rangle = 2\,\mathrm{tr}(C^\top W^\top dW) + 2\,\mathrm{tr}(C^\top dW^\top W).$$

Using $\mathrm{tr}(C^\top dW^\top W) = \mathrm{tr}((WC)^\top dW)$ and symmetry of $C$ (since $W^\top W$ is symmetric) gives

$$dE = 4\,\mathrm{tr}((WC)^\top dW),$$

so $\nabla_W E(W) = 4WC$.    $\square$

**Lemma D.6** (Gradient of $R_0(W) = \|\mathrm{OffDiag}(W^\top W)\|_F$). *Let $R_0(W) := \|\mathrm{OffDiag}(W^\top W)\|_F$ and $C := \mathrm{OffDiag}(W^\top W)$. If $R_0(W) \neq 0$, then*

$$\nabla_W R_0(W) = \frac{2}{\|C\|_F}\, W\, C.$$

*Proof.* $R_0(W) = \sqrt{E(W)}$ with $E(W) = \|C\|_F^2$. By the chain rule,

$$\nabla_W R_0(W) = \frac{1}{2} E(W)^{-1/2} \nabla_W E(W) = \frac{1}{2}\frac{1}{\|C\|_F}(4WC) = \frac{2}{\|C\|_F}WC,$$

using Lemma D.5.    $\square$

## D.8   REGULARIZER YIELDS A RESTORING DRIFT

**Theorem D.7** (One-step decrease of $E(W)$ under its gradient). *Let $E(W) = \|\mathrm{OffDiag}(W^\top W)\|_F^2$ and consider*

$$W^+ = W - \eta\lambda \nabla_W E(W) = W - 4\eta\lambda\, W\, C, \quad C = \mathrm{OffDiag}(W^\top W).$$

*Then for sufficiently small $\eta\lambda$,*

$$E(W^+) = E(W) - \eta\lambda \|\nabla_W E(W)\|_F^2 + O((\eta\lambda)^2)\|\nabla_W E(W)\|_F^2,$$

*so the leading drift is strictly negative whenever $WC \neq 0$.*

*Proof.* Use a Taylor expansion of $E$ along $\Delta W := -\eta\lambda \nabla E(W)$:

$$E(W^+) = E(W) + \langle \nabla E(W), \Delta W\rangle + O(\|\Delta W\|_F^2) = E(W) - \eta\lambda\|\nabla E(W)\|_F^2 + O((\eta\lambda)^2)\|\nabla E(W)\|_F^2.$$

   $\square$

**Connection to $R_0$.** Since $R_0 = \sqrt{E}$, the same conclusion holds for the unsquared penalty up to a rescaling by $\|C\|_F^{-1}$ via Lemma D.6.

### D.9 ADAMW VS. MUON: WHICH GRAM TERM DRIVES OFF-DIAGONALS?

**Theorem D.8** (Muon suppresses off-diagonal quadratic injection). *Let $W^+ = W + \Delta W$ and $G = W^\top W$. Assume a Muon-like update $\Delta W = -\eta Q$ with*

$$Q^\top Q = cI + E, \qquad c > 0,$$

*where $E$ is small (in Frobenius or operator norm). Then*

$$\mathrm{OffDiag}(\Delta W^\top \Delta W) = \eta^2 \, \mathrm{OffDiag}(E),$$

*so the quadratic term in Lemma D.3 contributes only $O(\eta^2 \|E\|)$ to off-diagonal Gram dynamics.*

*Proof.* Compute $\Delta W^\top \Delta W = \eta^2 Q^\top Q = \eta^2(cI + E)$ and note that $\mathrm{OffDiag}(cI) = 0$. $\qquad \square$

**Interpretation** Muon suppresses off-diagonal growth through $\Delta G^{(2)}$; remaining growth must come from cross-term drift $\Delta G^{(1)}$.

## E HYPERPARAMETERS

### E.1 OPTIMIZATION HYPERPARAMETERS

**Common training setup.** We follow the FineWeb(-Edu) preprocessing and learning-rate schedule of Ahn et al. (2025): a constant learning rate with a **10% linear cooldown**, and (when enabled) a **10% warmup** for AdamW. In the multi-scale runs, we fix the **global batch in tokens** (rather than sequences), using **1.0M tokens/step** as in Ahn et al. (2025). We also follow compute-optimal scaling guidance ("Chinchilla"), training to a token budget on the order of $\sim 20\times$ parameters (Hoffmann et al., 2022).

**AdamW hyperparameters.** Unless stated otherwise, AdamW uses $(\beta_1, \beta_2) = (0.9, 0.95)$ and $\lambda_{\mathrm{wd}} = 0.01$. For the three model scales $\{160\mathrm{M}, 405\mathrm{M}, 1.4\mathrm{B}\}$, we use the learning rates reported/selected in Ahn et al. (2025):

$$\eta_{\mathrm{AdamW}} \in \{0.002, \ 0.0016, \ 0.0012\} \quad \text{(for 160M, 405M, 1.4B respectively)}.$$

**Muon hyperparameters.** For Muon, we use momentum $\mu = 0.95$, $\lambda_{\mathrm{wd}} = 0.01$, and a single learning rate

$$\eta_{\mathrm{Muon}} = 0.01$$

shared across all three model scales, following the multi-scale setting of Ahn et al. (2025). (Non-matrix parameters, when present, are updated with the same scalar-parameter optimizer configuration as in Ahn et al. (2025).)

### E.2 DATA HYPERPARAMETERS

**Tokens per step (explicit).** If the global batch contains $B$ sequences of length $T$, then each step processes

$$N_{\mathrm{tok/step}} = B\,T.$$

In the multi-scale configuration of Ahn et al. (2025), this is set to $N_{\mathrm{tok/step}} = $ **1.0M tokens/step**.

### E.3 MODEL HYPERPARAMETER

**Model scales.** We train GPT-style decoder-only Transformers at three parameter scales: 160M, 405M, and 1.4B. We use $(L, d) = (12, 768)$, $(24, 1024)$, and $(24, 2048)$ respectively. Following Ahn et al. (2025), we set the number of attention heads to

$$n_{\mathrm{head}} \in \{6, \ 32, \ 32\} \quad \text{for } \{160\mathrm{M}, \ 405\mathrm{M}, \ 1.4\mathrm{B}\}\,,$$

and unless stated otherwise we fix the context length to $T$ across scales, consistent with a compute-optimal (Chinchilla-style) training perspective (Hoffmann et al., 2022).

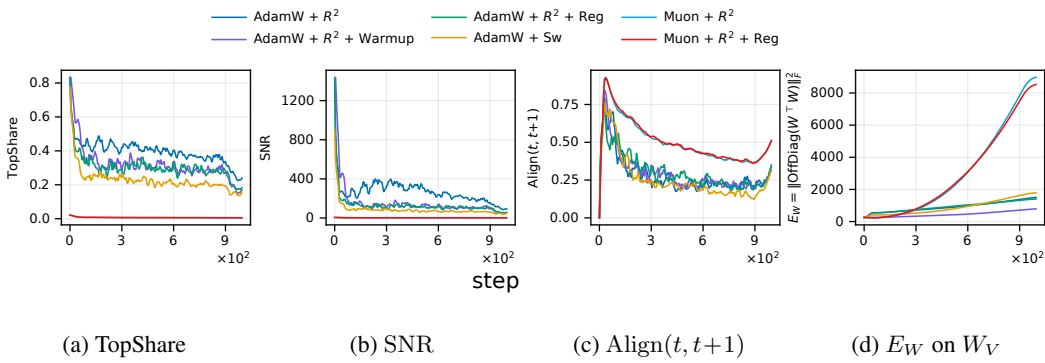

Figure 4: **Update-geometry and Gram-correlation at $W_V$ (layer-averaged).** Early-phase diagnostics of the effective update $\Delta W_V$ across training configurations: (a) temporal alignment $\text{Align}_V(t, t+1)$, (b) fluctuation anisotropy $\text{TopShare}_V$, (c) update coherence $\text{SNR}_V$, and (d) off-diagonal Gram energy $\|\text{OffDiag}(W_V^\top W_V)\|_F^2$.

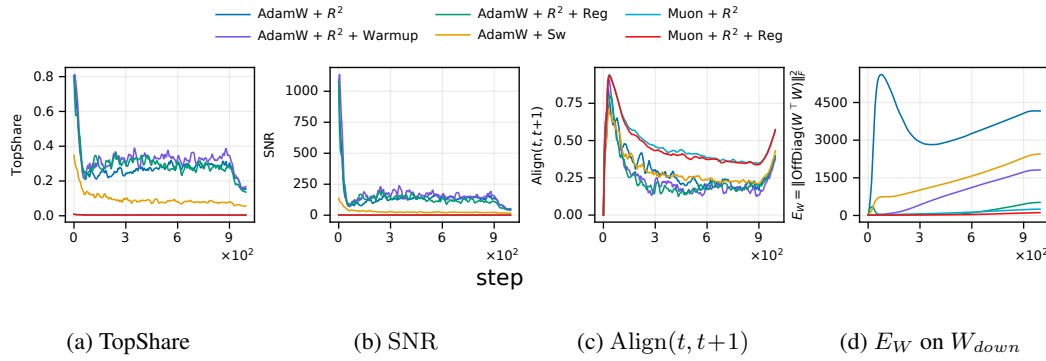

Figure 5: **Update-geometry and Gram-correlation at $W_{down}$ (layer-averaged).** Early-phase diagnostics of the effective update $\Delta W_{down}$ across training configurations: (a) temporal alignment $\text{Align}_{down}(t, t+1)$, (b) fluctuation anisotropy $\text{TopShare}_{down}$, (c) update coherence $\text{SNR}_{down}$, and (d) off-diagonal Gram energy $\|\text{OffDiag}(W_{down}^\top W_{down})\|_F^2$.

**MLP width and parameter-matching across activations.** To compare ReLU$^2$ and SwiGLU fairly, we match the MLP parameter budget by adjusting the expansion width $d_{\text{ff}}$. For ReLU$^2$ MLPs we use the standard two-projection form with expansion factor $4\times$, $d_{\text{ff}} = 4d$, with weights $W_{\text{up}} \in \mathbb{R}^{d_{\text{ff}} \times d}$ and $W_{\text{down}} \in \mathbb{R}^{d \times d_{\text{ff}}}$. For SwiGLU we use a gated MLP with two input projections (gate and non-gate) and one down projection, and set $d_{\text{ff}} = \frac{8}{3}d$ so that the dominant MLP parameter count matches ReLU$^2$.

### E.4    REGULARISER HYPERPARAMETERS

**Regularizer schedule and strength.** We apply the targeted off-diagonal Gram regularizer only during an early window of training: the regularization horizon is set to $t_{\text{reg}} = 0.1\, T_{\text{train}}$ (i.e., the first 10% of total steps), after which the regularizer is disabled. We use $\lambda = 10^{-3}$ for the 160M model and $\lambda = 10^{-4}$ for the 405M and 1.4B models.

## F    ADDITIONAL RESULTS

### F.1    GRADIENT PROFILES

### F.2    RESULTS USING SWIGLU

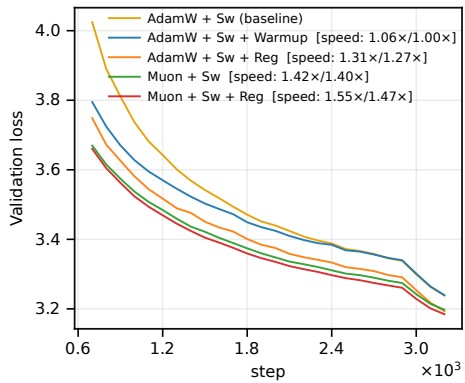

Figure 6: **Validation loss under SwiGLU with optimizer- and geometry-level interventions.** Curves show AdamW (baseline), AdamW+Warmup, AdamW+Reg, Muon, and Muon+Reg. The targeted Gram regularizer improves both AdamW and Muon, while Muon outperforms AdamW with/without warmup; combining Muon with the regularizer yields an additional improvement. Reported speeds are at two fixed loss targets, $L^\star = \{3.50, 3.35\}$

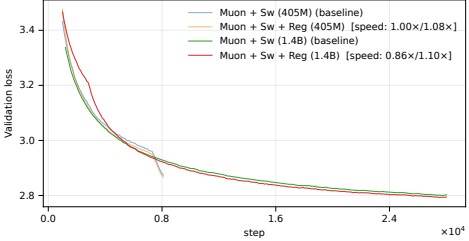

Figure 7: **Scale robustness of the intervention under Muon (SwiGLU).** Validation-loss overlays for Muon (baseline) and Muon+Reg at 405M and 1.4B parameters. The same early-phase regularizer consistently improves validation loss at both scales, while the larger model reaches a lower loss overall. Reported speeds are measured at two fixed loss targets: $L^\star \in \{3.25, 3.0\}$ for the 405M model and $L^\star \in \{3.10, 3.85\}$ for the 1.4B model.

