# OpenReview forum: "Learning in Transformers under Spectral Constraints"
_ICLR.cc/2026/Workshop/GRaM — ICLR 2026 Workshop GRaM Poster_

### Official Review · Reviewer_rHne · 2026-02-10

**Rating:** 6
**Confidence:** 3

**Review:**

**Overview** This paper observes that low-rank collapse in transformers is highly localized. This phenomenon causes the effective weight updates produced by AdamW to be low-rank. Assuming these updates are temporally aligned, it implies that the updates lie within a small subspace, which slows down optimization. The Muon optimizer helps mitigate this issue by orthogonalizing the effective updates.
By explicitly expressing the Gram matrix update in terms of the effective updates, the authors argue that Muon can only minimize the positive semi-definite (PSD) component. The off-diagonal terms can still grow, driven by the first term, leading to an increase in these off-diagonal values. To address this, the authors propose adding a regularization term that directly penalizes the off-diagonal elements during early training stage. The experiences show good results.

**Strengths**:
1. The regularisation term makes perfect sense for AdamW.
2. Good insight on localised rank collapse.
3. Strong experimental results to support the claim.

**Weakness/Questions**:
* The analysis explaining why this regularizer benefits Muon is rather weak. I don’t understand why the drift term would cause a significant accumulation in the off-diagonal terms. I would expect this effect to be quite small, since each effective update is approximately full rank. I would like to see a more detailed analysis supporting this.
* Even the off diagonal term is growing, the update returned by Muon would still be enough to escape the subspace update as it is always full-rank. Could the author provide more insight on why this helps?

If the author provides the above analysis, I think the paper is good for acceptance.


**Relevance to topics listed in GRaM call for papers:** Yes

**Originality and novelty:** Yes

**Technical soundness of method:** Yes

**Clarity in writing and organization of the paper** The paper is concise, the mathematical formulation is clear, and it respects the page limit.

**For the Proceedings track:** N/A

**Double-blind reviewing:** No violations of anonymity were found.

**Use of LLMs:** The text is technical and precise; there are no signs of excessive or improper LLM generation.

**Pmlr Suitability:**

Yes

---

### Official Review · Reviewer_wdtW · 2026-02-24

**Rating:** 6
**Confidence:** 3

**Review:**

**Summary:**  This paper identifies a spectral pathology in early Transformer pretraining: a small subset of weight matrices ($W_V$, $W_O$, $W_{down}$) develop severe spectral spikiness; rapid growth of the top singular value relative to the bulk; leading to ill-conditioning and slow convergence. The authors propose a "runaway correlated-update" mechanism to explain why these specific matrices are affected, and introduce a minimal early-phase off-diagonal Gram penalty applied only to the affected matrices for the first 10% of training.

**Strengths:** 1) Applying regularization only to three matrix types and only for the early training is simple and avoids the overhead of changing the optimizer globally. The fact that it helps both AdamW and Muon is a nice result. 2) The finding that early rank collapse concentrates in some layers, is a useful and actionable insight for practitioners.

**Weaknesses:** 1) The paper does not adequately discuss whether the intervention would matter at the scales where pretraining cost actually justifies optimization research (7B+). This weakens the practical contribution and all experiments use a relatively small models by current standards. 2) The flow of the paper seems like lots of details are taken out to be placed in the appendix and it is sometimes hard to understand their claims in first attempt.

**Pmlr Suitability:**

NA

---

### Meta-Review · Area_Chair_VosF · 2026-02-26

**Decision:**

Accept

**Metareview:**

The paper studies a spectral pathology in early transformer training. Reviewers appreciate the simplicity of method, strong experimental results, and useful insights. However, they also raise concerns regarding whether the proposed method brings advantage at scale, as well as further analysis of why the method benefits Muon. Overall, I recommend acceptance and encourage the authors to incorporate reviewers’ feedbacks in the final version.

**Relevance To Proceedings:**

Tiny paper — does not apply

**Relevance To Workshop:**

Yes — suitable for GRaM

---

### Decision · Program_Chairs · 2026-03-02

Accept (Poster)